# Charge transfer drives anomalous phase transition in ceria

He Zhu[1,2], Chao Yang[3], Qiang Li[1,2], Yang Ren[4], Joerg C. Neuefeind [5], Lin Gu [6], Huibiao Liu[7], Longlong Fan[1,2], Jun Chen [1,2], Jinxia Deng[1,2], Na Wang[1,2], Jiawang Hong[3] & Xianran Xing[1,2]

Ceria has conventionally been thought to have a cubic fluorite structure with stable geometric and electronic properties over a wide temperature range. Here we report a reversible tetragonal ($P4_2/nmc$) to cubic ($Fm$-$3m$) phase transition in nanosized ceria, which triggers negative thermal expansion in the temperature range of $-25\,°C$–$75\,°C$. Local structure investigations using neutron pair distribution function and Raman scatterings reveal that the tetragonal phase involves a continuous displacement of $O^{2-}$ anions along the fourfold axis, while the first-principles calculations clearly show oxygen vacancies play a pivotal role in stabilizing the tetragonal ceria. Further experiments provide evidence of a charge transfer between oxygen vacancies and $4f$ orbitals in ceria, which is inferred to be the mechanism behind this anomalous phase transition.

[1] Beijing Advanced Innovation Center for Materials Genome Engineering, Department of Physical Chemistry, University of Science and Technology Beijing, Beijing 100083, China. [2] State Key Lab for Advanced Metals and Materials, University of Science and Technology Beijing, Beijing 100083, China. [3] School of Aerospace Engineering, Beijing Institute of Technology, Beijing 100081, China. [4] X-Ray Science Division, Argonne National Laboratory, Argonne, IL 60439, USA. [5] Chemical and Engineering Materials Division, Oak Ridge National Laboratory, Oak Ridge, TN 37831, USA. [6] Beijing National Laboratory for Condensed Matter Physics, Chinese Academy of Science, Beijing 100190, China. [7] Institute of Chemistry, Chinese Academy of Sciences, Beijing 100190, China. Correspondence and requests for materials should be addressed to J.H. (email: hongjw@bit.edu.cn) or to X.X. (email: xing@ustb.edu.cn)

Ceria (CeO$_2$), which serves as structural stabilizer and electronic promoter, is one of the most important functional materials for a number of applications, such as catalysts for various catalytic reactions, solid oxide fuel cell (SOFC), biomedicine, etc[1–4]. The success of these applications requires a fundamental understanding of the unique property of ceria at the nanoscale, which is governed mostly by its short-ranged crystal structure (the so-called structure-property relationship). For nanosized ceria, the high oxygen storage capacity (OSC)[1–3,5], i.e., the excellent ability of storing and releasing oxygen, enables the lattice to tolerate considerable amount of oxygen vacancies and Ce$^{3+}$, giving rise to a more distorted lattice and profound orbital electronic dispersion than in the bulk. This makes the geometric and electronic structures of nanosized ceria a controversial topic, and conclusive experimental evidence of local structure is still missing to date. It has been suggested that nanoscale ceria preserves the fluorite structure even under strong oxygen deficient conditions[6], whereas there is a contrary view that a cubic form of Ce$_2$O$_3$ (C-type) might coexist in the fluorite lattice[7]. As a result, determining the nanostructure of ceria with complex symmetries is much desired, which is a challenging task.

Charge transfer related to oxygen vacancies during oxidation and reduction states plays a significant role in activating catalytic sites[8], and also in promoting electronic transportation for SOFC[9]. The description of electronic dispersion around defects is a challenge for ceria-based applications with improved efficiency. To this end, computational attempts have been made on the localization of the electrons left upon oxygen removal[10], but the employed models could be quite different from the ceria nanoparticles, and the temperature effect is another gap between fundamental studies and practical applications. Detailed experimental insights into the electronic structures of defects and their coupling with the local structures would therefore offer a significant step forward.

In the present study, we provide detailed experimental and computational insight into the nanostructure of ceria, revealing a charge-transfer-induced tetragonal-cubic phase transition promoted by oxygen vacancies. Comprehensive methods, including scanning transmission electron microscopy (STEM), X-ray diffraction (XRD), neutron pair distribution function (nPDF) and Raman spectroscopy, have been applied to probe the multiscale fine structures of nanosized ceria. In addition, the electronic structure coupled with defects has been revealed by a combination of electron paramagnetic resonance (EPR), X-ray photoelectron spectroscopy (XPS) and current–voltage (I–V) measurements. First-principles calculations have been also performed on phonon softening simulations and oxygen-vacancy modeling, which offer a deeper understanding of the anomalous phase transition in nanosized ceria.

## Results

### Phasetransition-induced anomalous thermal expansion in CeO$_2$.

Ceria nanocrystals with different particle sizes, i.e. 5 nm, 9 nm, and 18 nm according to the transmission electron microscope (TEM) images (Supplementary Fig. 1), were prepared using soft chemical method[11]. To eliminate the extra effect on the lattice, the capping oleic layers were completely removed without grain growth by annealing in air at 350 °C, as indicated by Fourier transform infrared (FT-IR) spectrum (Supplementary Fig. 2) and TEM images (Supplementary Fig. 3). The thermal lattice evolutions of the prepared CeO$_2$ samples were extracted initially from XRD Rietveld refinements with cubic-fluorite model (Supplementary Fig. 4, Supplementary Table 1). As seen in Fig. 1a, bulk ceria shows a linear positive thermal expansion, with the coefficient of thermal expansion (CTE, $\alpha$) to be $(9.86 \pm 0.30) \times$

$10^{-6}$ °C$^{-1}$. As the particle sizes reduce to the nanoscale, the thermal expansions in the temperature range of −25 °C to 75 °C are significantly weakened, exhibiting distinct steps in the thermal expansion curves. Negative thermal expansion (NTE) was observed ($\alpha = (-4.46 \pm 0.76) \times 10^{-6}$ °C$^{-1}$) for the 5 nm ceria, which can be recognized by the shifts of the (1 1 1) diffraction peaks (Fig. 1b). Such NTE behavior indicates a structural break for the nanosized ceria, which, however, cannot be further determined by XRD. This is due to the broadening and overlapping of the Bragg peaks that significantly reduces the XRD resolution[12].

In the NTE temperature range, an apparent peak was observed in the specific heat capacity ($C_p$) curve for the 5 nm ceria (Fig. 1c), which implies that the NTE could be triggered by a phase transition. The entropy ($\Delta S$) of this phase transition was estimated by integrating ($C_p$–$C_{fit}$)/$T$, where $C_{fit}$ is the background extracting with a polynomial function to $C_p(T)$ between −150 °C and 150 °C (Fig. 1c inset). The value of the estimated $\Delta S$ is 0.03$R$, which is much smaller than the value $R\ln 2$ expected for a complete second-order phase transition[13]. This indicates a fact that the phase transition could be relevant to a local structural change.

### Oxygen vacancies in nanosized CeO$_2$.

In order to provide a visualized view of the local structure, STEM with annular bright-field detector (ABF-STEM) was carried out for 5 nm ceria (Fig. 1d). Along the [001] projection, individual cerium- and oxygen-atom columns appear alternately, and the atomic arrangement is close to the *Fm-3m* structure of ceria. Faint dark dots could be recognized at the oxygen sites, but the contrast of these dots fluctuates, which is firmly demonstrated by the inconsistent peak valleys of the oxygen intensity profile (Fig. 1d inset). The weakening of the oxygen contracts could be directly related to the low occupation for the specific oxygen sites. In addition, the X-ray photoelectron spectroscopy (XPS) results (Supplementary Fig. 5) also demonstrate that the amount of Ce$^{3+}$ and oxygen vacancies increases dramatically as the particle size reduced to the nanoscale. As a result, there exist dispersively distributed oxygen vacancies in the lattice of nanosized ceria, which might be the key clue for the further analysis of the NTE and the phase transition in nanosized ceria.

### Probing tetragonal distortion by nPDF.

CeO$_2$ adopting fluorite-type structure possesses eight equivalent O$^{2-}$ anions coordinated around each Ce$^{4+}$ cation at the corner of a cube (Fig. 2a inset). The local distortion of this oxygen-sublattice for 5 nm ceria has been investigated by nPDF. As seen in Fig. 2a, the distance of the nearest O–O atom-pair, which lies on the cube edge and corresponds to the second PDF peak, contracts as the temperature rises from −25 to 75 °C. Meanwhile, the distances of the nearest Ce–O pairs and the next-nearest O…O pairs have no significant change. The inconsistent thermal expansion behaviors between the first three atom-pair distances certainly break the fluorite symmetry, which verifies the phase transition in the 5 nm ceria.

The low-$r$ nPDF data (from 1.5 to 15 Å) of 5 nm ceria was initially adopted with the cubic-fluorite model. This fluorite model fits well to the data above 75 °C, but fails to describe the data below 75 °C, especially below −25 °C. Accordingly, the agree-factors ($R_w$) of the nPDF fittings increase remarkably as the temperature decreases from 75 °C to −25 °C (Supplementary Fig. 6). This result suggests that the local structure of the nanosized ceria differs from that of the bulk at low temperature. Given that the local phase transition involves a contraction of the nearest O–O atom-pair distance, two subgroups of the fluorite structure, *I/mmm* and *P*4$_2$/*nmc*, are selected from the feasible distorted structures with the symmetry as high as possible. From

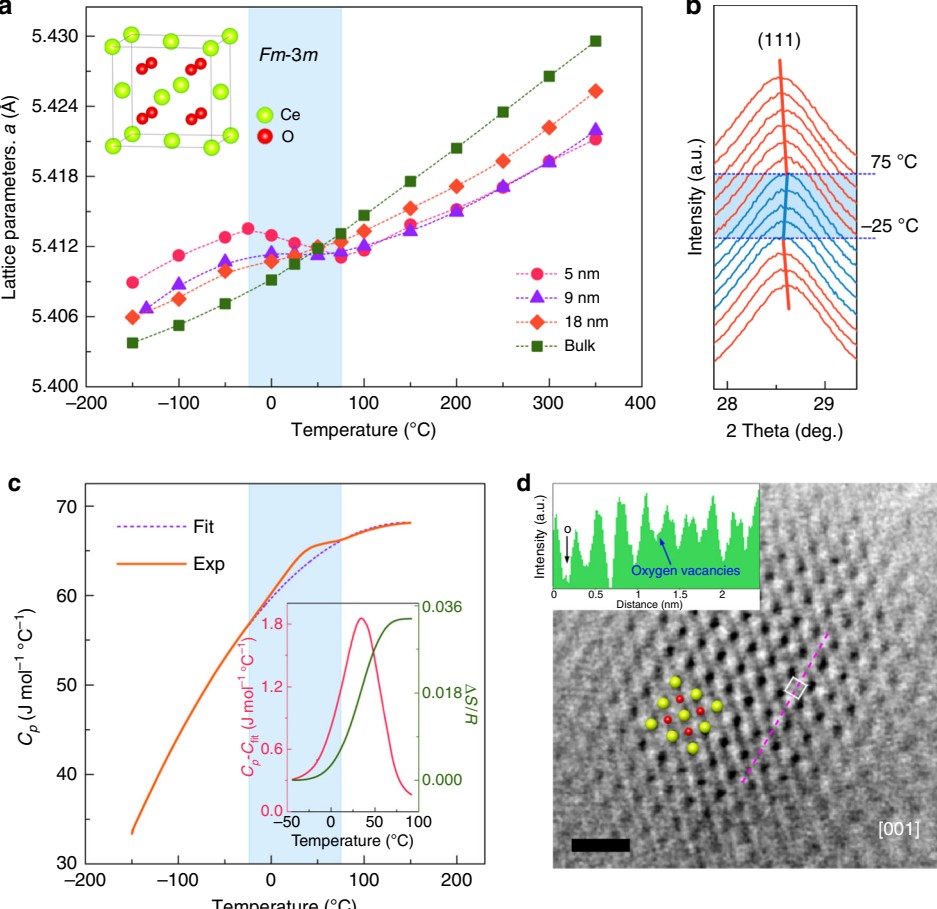

**Fig. 1** Phase transition-induced anomalous thermal expansion in nanosized ceria. **a** Temperature dependence of the lattice parameters extracting from X-ray diffraction Rietveld refinements for the $CeO_2$ in different sizes. The inset depicts the unit cell of $CeO_2$ with the space group *Fm-3m*. The errors are much smaller than the size of data symbols in the figure. The blue area emphasizes the temperature range of thermal expansion abnormity. **b** Comparison of the (1 1 1) diffraction peaks of 5 nm ceria data at different temperatures. The solid line depicts trend of the peak positions as the temperature changes. The blue area shows the temperature range of the negative thermal expansion. **c** Specific heat capacity of the 5 nm ceria measured from −150 °C to 150 °C. The red line in the inset is the heat capacity peak in the transition region excluding the fitted background (i.e., ($C_p$–$C_{fit}$)), and the green line shows the estimated entropy obtained by integrating ($C_p$–$C_{fit}$)/T. The blue area corresponds to the blue area in Fig. 1a. **d** The annular bright-field (ABF) image with a common tilt axis of [0 0 1] for 5 nm ceria. The scale bar is 1 nm. The inset shows the corresponding ABF in line profile acquired along the oxygen-atom columns (pink line in the figure). The arrow shows the oxygen vacancy site marked with hollow block

the refinement results (Fig. 2b), the tetragonal $P4_2/nmc$ model provides the best description of the data at low temperature. This tetragonal structure can be regarded as the shear strain of the oxygen sublattice distorted along fourfold axis (Fig. 2c inset), which leads to a smaller tetragonal unit cell than the cubic one. Based on the conversion relation between *Fm-3m* and $P4_2/nmc$ (Supplementary Fig. 7), the obtained tetragonal structure ($a_t$, $c_t$) can be expanded to a cubic-like unit cell ($a = \sqrt{2}a_t$, $c = c_t$). This expanded unit cell has been adopted in the present study to describe the tetragonal phase, so as to unify the tetragonal lattice constants towards the cubic structure.

The PDF fitting was carried out with $P4_2/nmc$ model over the entire temperature range to obtain continuous structural evolution (Supplementary Table 2). During the tetragonal-cubic phase transition from −25 °C to 75 °C, c-axis contracts towards the length of a-axis ($\sqrt{2}a_t$), while the oxygen coordinate along c-axis ($O_z$) approaches to 0.75 (Supplementary Fig. 8a). This leads to a continuous shift of $O^{2-}$ anions with almost 0.04 Å and also a volume contraction of 1.3‰ excluding the general thermal expansion (Supplementary Fig. 8b). It is worth mentioning that the PDF refinement didn't show a significant improvement with two-phase fitting with both cubic and tetragonal models. As a

consequence, the 5 nm ceria should be regarded as single tetragonal structure at low temperature. In addition, the observed tetragonal-cubic phase transition is reversible according to the XRD results under thermal cycling (Supplementary Fig. 9), indicating that the tetragonal structure is a stable phase other than an unstable reconstruction induced by short-range effect like polar termination[14]. The driving force of the $O^{2-}$ displacement could be the phonon condensation described later in our first-principles calculations.

**Lattice dynamics of the phase transition.** Raman scattering measurements supply more evidences to confirm this structural phase transition by determining the phonon vibrations. As shown in Fig. 3a, the main band at ~465 cm$^{-1}$ is the only allowed triply degenerate Raman mode ($F_{2g}$) of fluorite structure[15], which can be regarded as a breathing mode of the $O^{2-}$ anions around each cation (Supplementary Fig. 10a). The band at ~600 cm$^{-1}$ (D-band) corresponds to non-degenerate LO mode arising from relaxation of symmetry rules, which is linked to oxygen vacancies in ceria lattice[16]. Note that the phonon mode at ~276 cm$^{-1}$, however, is detected, which is correlated with the opposite

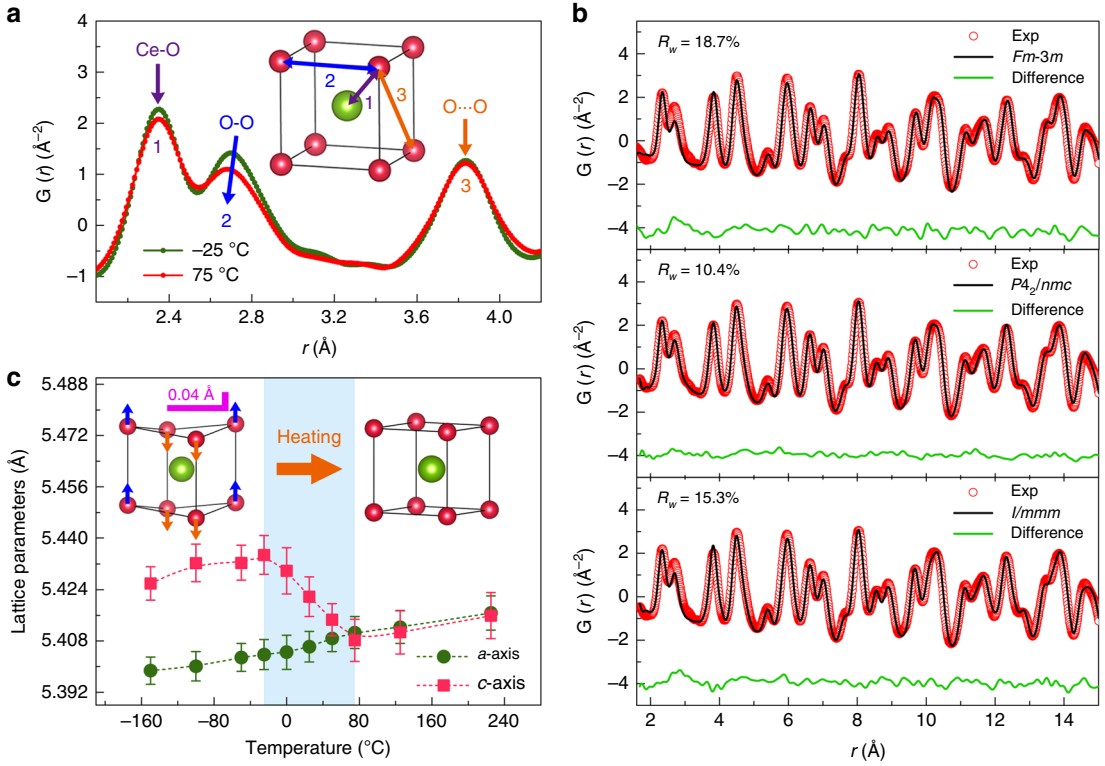

**Fig. 2** Local structural evolution of the phase transition. **a** Typical PDF Spectra of 5 nm ceria in the distance from 2 to 4.2 Å at the temperature of −25 °C and 75 °C. The inset is a schematic diagram of the first three atom-pairs in the Ce–O cube, which corresponds to the first three PDF peaks. The arrows show the trends of the peak shifts with temperature rising. **b** Examples of short-$r$ PDF refinements for the data taken at −25 °C with different structural models. **c** Temperature dependence of the lattice parameters extracted from the low-$r$ PDF refinements of 5 nm ceria samples. The inset is the schematic diagram of the local symmetric transition occurring with temperature rising in nanosized ceria. The blue area highlights the temperature range of the phase transition. The errors are estimated based on the standard deviations of least-squares fitting given by the PDFgui software

vibrations of Ce and O atoms along $c$-axis (Supplementary Fig. 10b). This mode is Raman-silence for the fluorite structure but becomes active for the tetragonal structure. Therefore it could be regarded as a signature of the tetragonal phase. As can be seen in the Fig. 3a inset, the integral area of this characteristic peak continuously decreases and disappears with temperature approaching to 200 °C, where the cubic phase is dominant.

The lattice dynamics, as well as the formation of intrinsic defect of nanosized ceria, has been also investigated using the density functional theory (DFT) with the generalized gradient approximation corrected for on-site Coulombic interactions (GGA + U, ref. [10]). The $U$-values of $Ce_{4f}$ and $O_{2p}$ states were set to 5 eV and 5.5 eV, respectively, in order to overcome the self-interaction error (SIE) in the reduced $CeO_2$ system[17] (see Methods for details). Considering the expansion of $c$-axis in the tetragonal structure, one of the axes in cubic structure was stretched as $c$-axis with the oxygen atoms at high-symmetry positions, then the phonon dispersion was calculated with VASP[18] and Phonopy[19]. As shown in Fig. 3b, there is a phonon branch with imaginary frequency at the Brillouin zone boundary M point (0.5, 0.5, −0.5), indicating that the structure is unstable due to the introduced strain along $c$-axis. This soft mode represents the vibration of the neighboring oxygen chains in opposite direction (Fig. 3c inset), exactly the same as the tetragonal distortion observed experimentally. The stable phonon dispersion of the $P4_2/nmc$ structure can be obtained by relaxing the structure with the modulation of the oxygen anions along the direction of the M-point soft phonons (Supplementary Fig. 11).

The condensation of the M-point soft mode leads to the distortion of the oxygen-sublattice towards the tetragonal $P4_2/nmc$

phase. We calculated the frozen phonon potential of this soft mode and obtained the double-well potential (Fig. 3c). As the temperature rises, the potential barrier between the tetragonal and cubic structures could be overcome, interpreting the tetragonal-cubic phase transition upon heating. The above phonon softening could arise from the surface stress induced by dispersively distributed oxygen vacancies in nano ceria (discussed later).

**Oxygen vacancy modeling in cubic and tetragonal lattice.** From the above experimental and calculated results, we suppose that the oxygen vacancies in nanosized ceria are likely to stabilize the tetragonal ceria, which has been modeled by the first-principles calculations. The $Fm$-$3m$ unit cell with one oxygen vacancy ($CeO_{1.75}$) was employed initially for the structural optimization. When a small perturbation along $c$ direction was applied to one of the oxygen atoms, this non-stoichiometric structure will be spontaneously relaxed to the $P4_2/nmc$ phase with 3.5 meV per atom lower than the cubic $CeO_{1.75}$ structure (Supplementary Fig. 12). This means the tetragonal structure becomes more stable when introducing oxygen vacancies in the lattice, which is consistent with our experimental results at low temperature. In such case, the conventional cubic structure could become a metastable state, which turns into the tetragonal phase with a slight atomic-site fluctuation (perturbation). For the nanosized ceria, such fluctuation is frequently observed due to the short-range coherence of the lattice, giving rise to the tetragonal structure at low temperature.

**Charge transfer in defective nano CeO₂.** Theoretically, two $Ce^{3+}$ sites could be reduced by the residual electrons upon one neutron

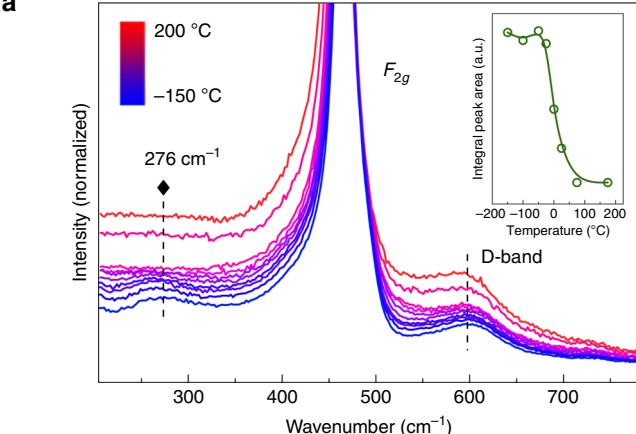

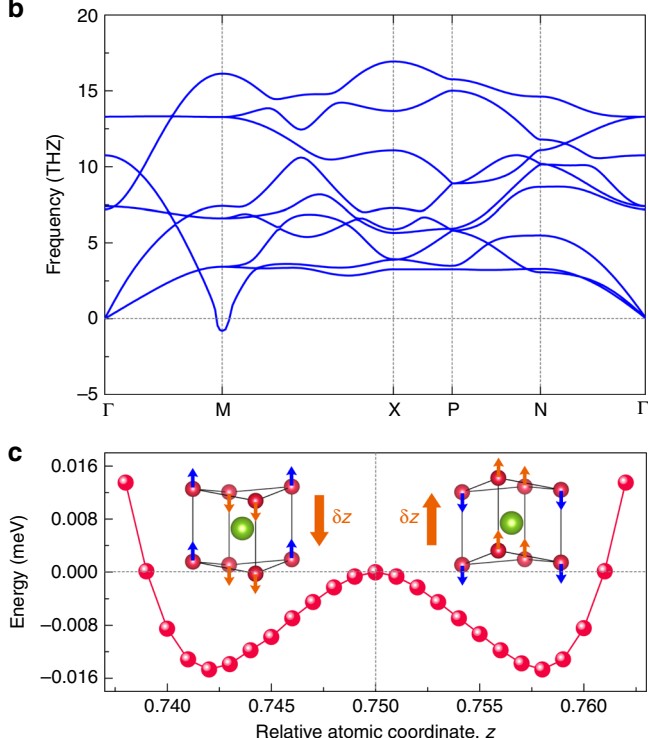

**Fig. 3** The phonon characters of nanosized ceria. **a** Raman spectra of 5 nm $CeO_2$ in variation temperatures. The dash lines emphasize the peak position of the two new vibration modes apart from the $F_{2g}$. The inset depicts the temperature dependence of the areas of the tetragonal characteristic peaks obtained by Gaussian fitting. **b** The phonon dispersion of the elongate structure by stretching the lattice along c-axis. The high-symmetry points of the Brillouin zone are denoted as Γ (0 0 0), M (0.5 0.5 −0.5), X (0 0 0.5), P (0.25 0.25 0.25), N (0 0.5 0). **c** The systematic energies as a function of the oxygen displacement along the vibration of the phonon at the M point. The insets are the schematic diagrams of the distortion of the oxygen sublattice corresponding to the double-well potential

oxygen removal[10]. However, the electron paramagnetic resonance (EPR) spectra of the 5 nm sample show that the residual electrons could be localized in oxygen vacancies for the tetragonal ceria, and an electron transfer occurs accompanied with the phase transition. As seen in Fig. 4a, the symmetric signal at $g = 2.003$ is assigned to the unpaired electrons trapped in oxygen vacancies (paramagnetic defect with excess spins, $V_o^-$)[20]; whereas, the axial signals with $g_\perp = 1.967$ and $g_{//} = 1.947$ are assigned to the paramagnetic $Ce^{3+}$ sites with unpaired $f$ electrons[21]. During the phase

transition upon heating, a charge transfer occurs from the ($V_o^-$ s) of tetragonal ceria to the Ce $f$ orbitals of cubic ceria (Fig. 4b), suggesting that the charge states of defects in the energy gap could play a key role in the stability of different configurations. In addition, such charge transfer between tetragonal and cubic phase has been also verified by our DFT calculation of spin charge density (Supplementary Fig. 13).

The electrons localized in defects could serve as the mobile charge carriers that increase the conductivity[22], which motivates us to concern the electric transportation property. The I–V measurements of the film coated with 5 nm ceria particles clearly show this scenario with temperature (Fig. 4c, d, see Methods for technical details). At an applied voltage of 10 V, the relative current in tetragonal ceria at liquid nitrogen temperature is about two orders of magnitude higher than that in cubic ceria at 100 °C (Fig. 4d), although the conductivity of bulk ceria exhibit no obvious change (Supplementary Fig. 14). The nonlinear of the I–V curves of 5 nm ceria are due to the Schottky contact between the ceria film and the Au electrodes[23]. This switch of conductivity between tetragonal and cubic ceria indicates the charge transfer during the phase transition, and also makes nanosized ceria as candidates for extended electrical applications.

## Discussion

From the above experiments and simulations, a conclusion could be drawn that the oxygen vacancies in the lattice play a key role in stabilizing the tetragonal ceria. Our ABF image clearly shows that the distribution of the oxygen vacancies in nanosized ceria is dispersive (Fig. 1c). Such inhomogeneity of the point defects gives rise to surface stress (the so-called chemically induced stress, refs [24,25]), and thereby softens the phonon mode (Fig. 3b). As the particle reduces to nanoscale, the amount of $Ce^{3+}$ increases dramatically (see the XPS results in Supplementary Fig. 5), leading to a more significant stress to drive the phase transition in nanosized ceria. On the other hand, the electronic structures around oxygen vacancies are different in the tetragonal and cubic ceria. The residual electrons from oxygen vacancies tend to be trapped in the vacancies for the tetragonal ceria but occupy Ce $f$ orbitals ($Ce^{3+}$) for the cubic ceria, and the charge transfer occurs during the tetragonal-cubic phase transition (Fig. 4a, b). Similar with the carrier-injection-induced phase transition in α-$SiO_2$[26], the observed charge transfer associated with defects could be the mechanism in electronic level behind the phonon softening and the phase transition in nanosized ceria.

In summary, the tetragonal phase in nanosized ceria differs from the structure of bulk non-stoichiometric $CeO_{2-\delta}$[27], indicating the existence of critical size for the phonon condensation and defect-induced charge transfer. Also, the size-tuning softening feature was found in phonon mode, which opens a window to predict the complex structural change in nanomaterials. In addition to the scientific interest, the phase transition and the charge transfer in nanosized ceria are important from the catalytic point of view, as it occurs within the moderate temperature range below 100 °C. The high concentration of electron donors and improved conductivity for tetragonal ceria might provide a unique perspective for understanding the high OSC and catalytic reactivity of nanosized ceria.

## Methods

**Non-hydrolytic sol-gel synthesis of 5 nm ceria**. Cerium (III) nitrate hexahydrate (1.7 g, analytical grade) was added to oleylamine (20 mL, 16.26 g, technical grade) at room temperature. The resulting solution was heated to 100 °C under $N_2$ atmosphere, and a homogeneous, clear, reddish brown solution formed. Diphenyl ether (2 mL, 2.1 g, analytical grade) was injected into the solution at 100 °C, and the observed temperature increased to 120 °C due to the reaction between cerium nitrate and diphenyl ether. The resulting mixture was heated to 320 °C and aged at that temperature for 2 h to give a dark red colloidal solution. Ethanol (100 mL) was

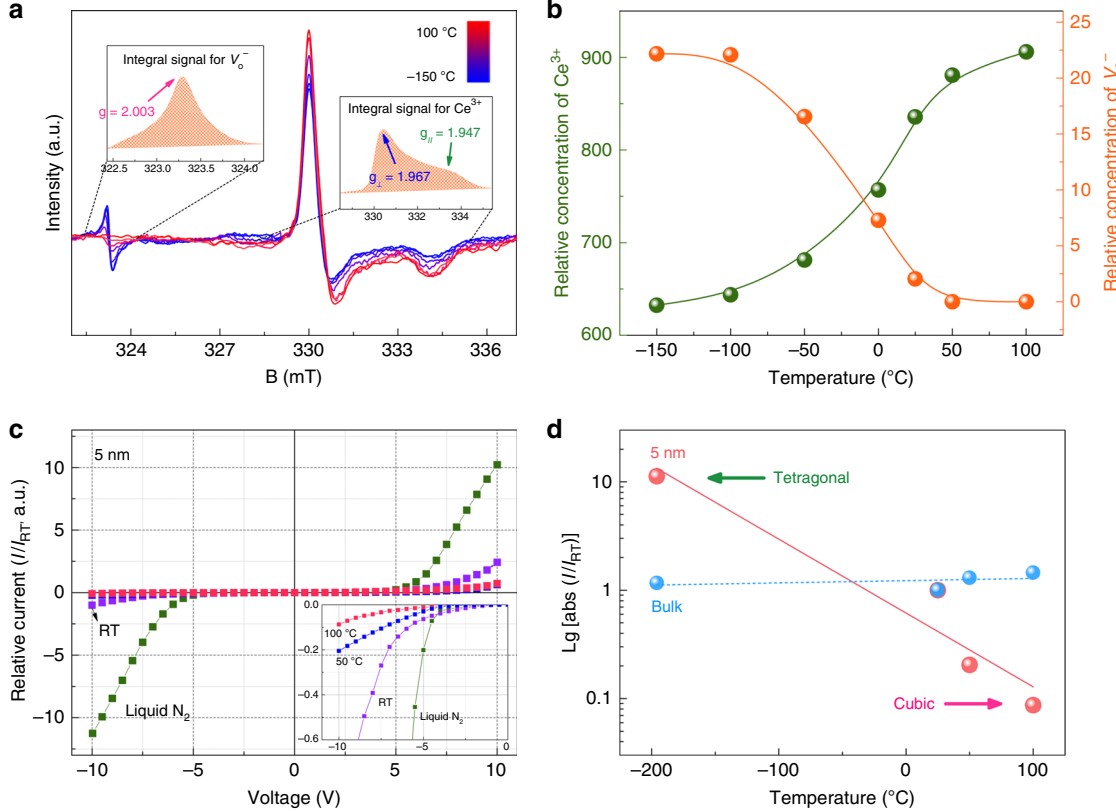

**Fig. 4** Enhanced conductivity in tetragonal ceria results from electron transfer during the phase transition. **a** Temperature variation of electron paramagnetic resonance (EPR) spectra of 5 nm ceria. The left inset is an example of the integral signal for the paramagnetic defect ($V_o^-$) and the right inset is an example of the integral signal for $Ce^{3+}$. The corresponding integration ranges are depicted with black dashed lines. **b** The relative concentration of the $Ce^{3+}$ and the $V_o^-$ species as a function of temperature extracted from integrating the EPR signals. **c** The current–voltage ($I$–$V$) curves for the 5 nm sample under varying temperature conditions. All the curves are normalized by the room temperature current ($I_{RT}$) at −10 V. The inset depicts the expanded low current region. **d** Temperature dependence of the log-scaled relative currents, $\lg\left|\frac{I}{I_{RT}}\right|$, with the applied voltage of 10 V for both 5 nm and bulk sample. The arrows correlate to the local structures of 5 nm ceria

added to precipitate the ceria nanocrystals. The precipitate was retrieved by centrifugation to give white–brown ceria nanocrystals. The nanocrystals could be dispersible in nonpolar solvent (toluene, hexane, etc.).

In addition, the bulk ceria was obtained by calcining the 5 nm nanoparticles in air at 1100 °C.

**Hydrothermal synthesis of 9 nm and 18 nm ceria**. Cerium nitrate aqueous (15 mL, 16.7 mmol $L^{-1}$) was transferred to a 45 mL Teflon-lined stainless-steel autoclave, and then toluene (15 mL), oleic acid (1.5 mL, technical grade), and *tert*-butylamine (0.15 mL, analytical grade) were added to the autoclave in the ambient environment without stirring, respectively. For 9 nm nanocubes, the sealed autoclave was transferred to a 150 °C oven and held there for 12 h, and then cooled to room temperature naturally. Whereas for the 18 nm nanocubes, the temperature should be 180 °C and the holding time should be 48 h. The crude solution containing the nanoparticles was precipitated with ethanol and further isolated by centrifugation in nonpolar solvent (toluene, hexane, etc).

**Removal of capping agents**. To eliminate the extra effects on the nanostructure, the capping oleic acid layers were completely oxidized without grain growth by annealing in air at 350 °C, which is confirmed using FT-IR and TEM (Supplementary Figs. 2–3).

**Sample characterizations**. TEM images were taken with JEOL JEM-2010 equipment operated at 200 kV. The ABF images were taken by an atomic-resolution analytical microscope (JEM-ARM 200 F) operating at 200 V. Samples for TEM and STEM-ABF were prepared by dropping dilute toluene dispersion of nanocrystals onto a carbon-coated copper grid and evaporated at ambient temperature. The FT-IR measurements were performed with a Varian Excalibur 3100 spectrometer. The spectra of dried KBr plate were deducted as background to eliminate the extra signal of water and hydroxyl in KBr. The variable temperature Raman spectra were collected on LabRAM HR Evolution of HORIBA using laser with wavelength of 532 nm. The XPS Spectra were acquired on a Kratos Axis ultra-imaging spectrometer. The EPR measurements were conducted on a JES-FA200 X-

band spectrometer ($v \approx 9.06$ GHz) operating at a 100 kHz field modulation. For the $I$–$V$ measurements, 3 mg $CeO_2$ nanoparticles was initially dispersed in 5 ml $H_2O$, and the precursor solution was dropped on $SiO_2$/Si substrate (thickness of $SiO_2$: 300 nm). Then the substrate was rotated using a Holmarc spin coater. After each coating, Au electrodes were thermally deposited on the as-prepared $CeO_2$ film through a shadow mask with a channel width of 80 μm. The $I$–$V$ characteristics of the device was recorded in air using a Keithley 4200 SCS apparatus and a Micromanipulator 6150 probe station.

**Collection and analysis of XRD**. The variable temperature XRD patterns were collected with a low temperature attachment of PW 3040-X'Pert Pro diffractometer from PANalytical (Cu Kα). The configuration of the low temperature attachment is shown in Supplementary Fig. 15. The heating conducting grease was smeared between the stage and sample cell, and the thermocouple is made of Pt100. The heating rate was 5 °C per minute and the holding time was 20 min before the data collections. All the XRD measurements were carried out under vacuum. The diffraction patterns were analyzed by the Rietveld refinements with the Fullprof software. For the 5 nm ceria, the lattice constants extracted from variable temperature data have been calibrated by quartz ($SiO_2$) internal standard. Note that the difference of the lattice constants extracted with or without the internal standard is subtle, which indicates the systematic error, derived from the change in sample height, has been corrected maximally through the Rietveld refinements.

**Collection and analysis of nPDF**. The neutron total scattering were collected on the nanoscale-ordered materials diffractometer (NOMAD) at the Spallation Neutron Source, Oak Ridge National Laboratory (SNS, ORNL). The $CeO_2$ powders with different grain sizes were loaded into a 2 mm quartz capillary sealed with a plastic stopper, with data acquisition time of 2 h for each sample. The background scattering from the empty container and instrument was subtracted. The data processing of PDF was done using the beamline-specific software. The experimental PDF was obtained by a Fourier transform of S(Q) up to a $Q_{max}$ of 31.4 Å. All experimental PDFs were analyzed using the PDFgui software package[28].

**Computational methods**. All the calculations in the present study were performed with Vienna ab initio simulation package (VASP)[18], using the Perdew–Burke–Ernzerhof (PBE) generalized gradient approximation (GGA) and the projector augmented wave (PAW) potential[29]. The energy cutoff of the plane-wave basis is 500 eV. It is known that standard DFT functionals are incapable of correctly modeling O-derived defect states due to the inherent self-interaction error (SIE)[10,17,30,31]. For the reduced $CeO_2$ system, i.e., $CeO_2$ that contains intrinsic O vacancies, such problem is acute for both $Ce_{4f}$ and $O_{2p}$ states[32,33]. To correct the SIE associated with the vacancies, $U\{Ce_{4f}\} = 5.0$ eV was applied to the Ce 4f states[34-37], while $U\{O_{2p}\} = 5.5$ eV, which is determined from a Koopmans-like fitting process[30], was applied to the O 2p states[17,32,33].

The first Brillouin zone is sampled with $4 \times 4 \times 4$ Monkhorst-Pack **k** mesh. The energy convergence threshold is set to $10^{-6}$ eV and the residual force on each atom is smaller than $10^{-3}$ eV per Å. In the calculation of the phonon dispersion, a $2 \times 2 \times 2$ supercell and a Monkhorst-Pack mesh of $4 \times 4 \times 4$ are used to calculate the second-order interatomic force constants within the finite displacement method, and then the phonon dispersion is obtained using the PHONOPY package[19]. All the phonon dispersions are calculated by taking LO-TO splitting into account, since the long-range Coulomb Interaction in $ZrO_2$, which has the similar phase transition from $P4_2/nmc$ to $Fm$-$3m$, cannot be neglected[38]. The spin charge density was calculated based on the previous research[39], for the tetragonal unit cell (using the experimental PDF results at $-150$ °C) and the cubic unit cell (using the experiment PDF results at 200 °C). One oxygen atom was removed for both tetragonal and cubic structure in conventional cells.

## Data availability

The data that support the findings of this study are available from the authors upon reasonable request, see author contributions for specific datasets.

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

## Acknowledgements

We thank K. Page and Q. Liu for the help of the nPDF measurements and the useful discussions, M. H. Quan for the insightful discussion of the TEM experiments. This work was supported by the National Natural Science Foundation of China (No. 21590793 and 21731001), the Program for Changjiang Scholars, the Innovative Research Team in University (IRT1207), and the Program of Introducing Talents of Discipline to Universities (B14003). J.H. acknowledges the support from the Thousand Young Talents Program of China and the National Science Foundation of China (Grant No. 11572040). Theoretical calculations were performed using resources of National Supercomputer Center in Guangzhou, which is supported by Special Program for Applied Research on Super Computation of the NSFC-Guangdong Joint Fund (the second phase) under Grant No. U1501501. This research used resources of the Advanced Photon Source, a U.S. Department of Energy (DOE) Office of Science User Facility operated for the DOE Office of Science by Argonne National Laboratory under Contract No. DE-AC02-06CH11357. A portion of this research also used resources at the Spallation Neutron Source, a DOE Office of Science User Facility operated by the Oak Ridge National Laboratory.

## Author contributions

H.Z., J.C., J.D., and X.X. conceived this study and designed the experiments. H.Z. and Q. L. conducted the experiments of XRD, TEM, specific heat capacity, Raman, EPR FT-IR, and XPS with analysis. L.G. conducted STEM with ABF detectors. H.Z., J.C.N., Y.R., and L.F. performed the nPDF measurements. J.H., C.Y., and N.W. designed and performed theoretical calculations. The I–V measurements were performed by H.L. H.Z., X.X., and J. H. wrote the manuscript with the assistance of all the co-authors.

**Additional information**

**Competing interests:** The authors declare no competing interests.

