## [Peer Review File · Nature Communications]

Reviewers' comments:

Reviewer #1 (Remarks to the Author):

The present manuscript „Charge transfer drives anomalous phase transition in ceria“ is well written and discusses exciting findings related to the technically—as described in the introduction—versatile cerium dioxide. To my knowledge, an unknown crystal structure of ceria has been identified by the authors and was causally related to nano-sized ceria including oxygen point defects. A wide range of spectroscopic characterization techniques (including EPR) has been applied and the degree of reduction was checked by state-of-the-art XPS. The mode triggering the phase transition from the cubic to a (lower-symmetry) tetragonal phase has been identified/characterized by means of DFT calculations on distorted cubic cell of CeO₂. Since no reduced ceria (involving O point defects) was applied the frequently used generalized-gradient-approximation after Perdew, Burke, and Ernzerhof (PBE exchange-correlation functional) can be applied. The very compact presentation of yet many spectroscopic details is very appealing. The described “phase transition as stress relief” is plausible. Of course, for theoreticians it would be interesting to know, whether the electrons in the O vacancy, creating an F-center is more or less stable than the Ce³⁺ (inducing strong relaxation effects and thus inducing stress) by applying the known but also discussed DFT+U approach using a U parameter of about 4 to 5 eV. However, it is also known that the stability of Ce³⁺ is likely to be overestimated by this approach. I see that this paper is certainly stimulating many interesting discussions in the oxide community and I am inclined to recommend publication as it stands.

Reviewer #2 (Remarks to the Author):

The manuscript deals with the phase transitions in CeO₂ and how they can be interpreted due to charge transfer effects. While the experiments are sound and the interest of CeO₂ is important I think that the justification particularly based on the Density Functional Theory modeling can be improved to a great extent. Therefore, I cannot recommend the manuscript for publication at the present stage.

At least a line on the computational details is required for the main text. This information already allows to calibrate the reliability of the computational approach.

The DFT calculations seem the minimum ad-hoc work. I do not understand why the authors do not provide the effect of vacancy formation from a computational point of view. This would help to understand if the origin of the tetragonal distortion is due to some particular arrangement of the generated vacancies. This study is a must to proceed further with the analysis. What is the role of the vacancies in the phonon dispersion?

The authors claim in page 8 that the electrons can be localized at the vacancy for the tetragonal ceria. This can actually be calculated through DFT with some degree of accuracy. As it is known it is quite speculative and the results seem to preliminary to be published in the present form.

What is the origin of the critical size observed for the transition?

The conductivity can also be analyzed in terms of the electronic structure of the defective ceria.

The relevance of Figure 1c needs to be clarified.

The inset in Figure 1d needs to be redrawn to improve readability.

The authors indicate that the phase transition
An extensive revision of the English is required.

Reviewer #3 (Remarks to the Author):

This manuscript describes a new phase of nanosized ceria stabilized by oxygen vacancies, and attributes the reversible phase transition to charge transfer. In addition, negative thermal expansion is reported over a 100 degree range due to this phase transition. As such, the topics are certainly of interest. The English is readable, however, it should be corrected before publication. As several authors reside in the United States, this should not be too difficult to achieve.

My concern with the manuscript in its current shape centers around the structural characterization of the nanosized ceria. As this is the central topic of interest, it is crucial that a number of issues are addressed/clarified before this paper can be further considered. I feel that important information is missing from this manuscript, and that some experiments need revisiting. These clarifications/result details will be necessary to truly evaluate whether this manuscript is appropriate for this journal.

The supplemental information should contain some more details. E.g., what exactly is the configuration for the "low temperature attachment" for the PXRD instrument? Is this a flat plate setup? If so - then variable temperature data ought to be collected with an internal standard to correct for sample height changes as a function of temperature. Note that this is especially important when dealing with small changes in lattice constants as in the current paper! The PDF data clearly show a shortening of nearest-neighbor O-O distances, but the absolute numbers could be affected considerably.

It is unclear whether the reported materials actually adopt a single phase, or whether a 2-phase mixture is formed. The authors clearly state that data up to 75 C require use of the tetragonal phase to get a good fit to the PDF data, but do not discuss whether all data can be accounted for exclusively by the tetragonal phase (e.g., the X-ray data should also be fittable with this phase!). Figure 1a shows lattice constants for a cubic phase extracted over the entire temperature range, and so does Table S1, implying that the cubic phase is observed at all temperatures. Table S2 seems to imply that the tetragonal phase is present at all temperatures up to 225 C. If both are correct/meaningful, then this should mean 2-phase coexistence. Of course, application of both models to the same data would then be necessary to determine relative amounts. Figure 2c shows tetragonal lattice constants at low temperature, and I presume cubic ones at higher T (this should be indicated somewhere!) - but the values for the tetragonal a parameters do not agree with the numbers in Table S2. Figure S8 claims to be displaying the a-parameter - yet once again, it is unclear whether this is the cubic or tetragonal parameter. Similarly, is Figure S6 displaying both cubic and tetragonal volumes? This is rather confusing, and makes it hard to truly evaluate the results presented. The authors need to clarify which phase is present at what temperature and refine/fit ALL data accordingly. E.g., if the material is not cubic at low T - then plotting a cubic lattice parameter as $f(T)$ at low T is not meaningful. If the material forms a 2-phase mixture, obviously major revisions will be necessary to adequately present the results.

Responses to Reviewers' comments

The point-by-point responses to the reviewers' comments are given below. All the corrections in the manuscript are highlighted in blue.

Responses to Reviewer #2

Q1: At least a line on the computational details is required for the main text. This information already allows to calibrate the reliability of the computational approach.

A1: Thank you for your suggestion. We have added more computational details in the main text.

Correction in the manuscript:

In order to investigate the lattice dynamics of nanosized ceria, the first-principle calculations, as implemented in Vienna ab initio simulation package (VASP), are performed using Perdew-Burke-Ernzerhof (PBE) generalized gradient approximation (GGA) and the projector augmented wave (PAW) potential with the plane cut-off of 500 eV and 4×4×4 Monkhorst-Pack k mesh (see the technical details in Methods in Supplementary Information).

Q2: The DFT calculations seem the minimum ad-hoc work. I do not understand why the authors do not provide the effect of vacancy formation from a computational point of view. This would help to understand if the origin of the tetragonal distortion is due to some particular arrangement of the generated vacancies. This study is a must to proceed further with the analysis. What is the role of the vacancies in the phonon

dispersion?

A2: This comment leads us to take a significant step forward to the tetragonal-stabilizing mechanism using the DFT + U methodology¹. First, we removed one oxygen atom (i.e. 12.5 % vacancies) from the $Fm-3m$ conventional cell of CeO_2 , and performed the structural optimization. No oxygen atom displacement was observed in this fully relaxed structure (marked as C_{vac} structure). However, when we induced a small perturbation along c direction to one of the oxygen atoms, this nonstoichiometric structure will be spontaneously relaxed to the $P4_2/nmc$ phase (marked as T_{vac} structure), with a slight stretch of c -axis and a small displacement of O along c -axis (the relaxed displacement pattern is the same as the experimental pattern). In addition, the total energy of the T_{vac} is lower than that of C_{vac} (Table R1). Hence, our DFT calculations show that the tetragonal structure becomes more stable when inducing oxygen vacancies, which is consistent with our experimental results. This calculation along with the related discussion is supplemented in the main text. The phonon dispersions for both C_{vac} and T_{vac} structure are calculated with VASP and Phonopy² (Fig. R1). Although the T_{vac} structure is more stable than C_{vac} , the distinction of their phonon dispersions is insignificant. We agree with the reviewer that the specific arrangement of the vacancies might make a greater difference to the phonon. However, this also leads to a tremendous computing workload (at least 36 configurations for $2 \times 2 \times 2$ supercell of $CeO_{1.75}$), which obviously lies outside the scope of the present study. We believe our experimental results and the phonon softening model, as well as the above DFT calculations with oxygen vacancy could

give a clear picture of the tetragonal-cubic phase transition in nanosized ceria.

Figure R1. The calculated phonon dispersions for the C_{vac} and T_{vac} unit cells.

Table R1: The calculated structural results of the C_{vac} and T_{vac} unit cells.

	a -axis (Å)	b -axis (Å)	c -axis (Å)	O_Z	Total energy (eV)
C_{vac}	5.51330	5.51330	5.51330	0.75	-60.055
T_{vac}	5.51466	5.51466	5.51468	0.752	-60.100

Supplemented in the manuscript:

From the above, we propose that the formation of oxygen vacancies plays a key role in stabilizing the tetragonal ceria, which has been also investigated by the first-principle calculations. The $Fm-3m$ unit cell with one oxygen vacancy ($CeO_{1.75}$) was employed for the structural optimization. When a small perturbation along c direction was applied to one of the oxygen atoms, this nonstoichiometric structure will be spontaneously relaxed to the $P4_2/nmc$ phase with 3.5meV/atom lower than

cubic $\text{CeO}_{1.75}$ structure (Fig. S13). Consequently, the tetragonal structure becomes more stable when inducing oxygen vacancies, which is consistent with our experimental results at low temperature. In such case, the conventional cubic structure could become a metastable state, which turns into the tetragonal phase with a slight atomic-site fluctuation (perturbation). For the nanosized ceria, such fluctuation is frequently observed due to the short-range coherence of the lattice, giving rise to the tetragonal structure at low temperature.

Added in Supplementary Materials:

Figure S13. The schematic diagram of the oxygen displacement for the relaxed tetragonal structure containing one oxygen vacancy.

Q3: The authors claim in page 8 that the electrons can be localized at the vacancy for the tetragonal ceria. This can actually be calculated through DFT with some degree of accuracy. As it is known it is quite speculative and the results seem to be preliminary to be published in the present form.

A3: Thank you for your suggestion. The paramagnetic defects in the tetragonal ceria

have been confirmed by the DFT calculation of spin charge density³. We found excess spin charge in the vacancy of tetragonal unit cell, evidencing the existence of the unpaired electrons localized at the vacancy (Fig. S14a). When the tetragonal ceria transforms to cubic ceria, the trapped electrons transfer to Ce 4*f* orbitals (Fig. S14b), which is consistent with the EPR result.

Correction in the manuscript:

During the phase transition upon heating, a charge transfer occurs from the (V_o^-s) of tetragonal ceria to the Ce *f* orbitals of cubic ceria (Fig. 4b), which has been also verified by our DFT calculation of spin charge density (Fig. S14).

Added in Supplementary Materials:

The spin charge density was calculated based on the previous research for the tetragonal unit cell (using the experimental PDF results at -150 °C) and the cubic unit cell (using the experiment PDF results at 200 °C). One oxygen atom was removed for both tetragonal and cubic structure in conventional cells.

Figure S14. The spin charge density of (a) tetragonal phase and (b) cubic phase with one oxygen vacancy in their unit cells.

Q4: What is the origin of the critical size observed for the transition?

A4: On the one hand, the defect concentration, which is highly correlated with the dimension of the nanoparticles, is the origin of the critical size for the phase transition. Our XPS results (Figure S12) demonstrated that the amount of oxygen vacancies increases dramatically when reducing the particle size. The inhomogeneous dispersion of the abundant vacancies gives rise to surface stress^{4, 5}, which softens the phonon mode to induce the phase transition. For the ceria with larger particle size, less significant stress is induced due to the declined defect concentration. Consequently, there could be a critical size (critical defect concentration) for ceria, above which the induced stress is not strong enough to soften the phonon, and thereby the charge

transfer and the phase transition is inhibited.

On the other hand, bulk non-stoichiometric $\text{CeO}_{2-\delta}$, which possesses high concentration of oxygen vacancies, has been found to adopt irreversible structural reconstruction instead of reversible phase transition⁶. Consequently, there exists another key factor, apart from the defect concentration, to trigger the phonon softening and charge transfer, giving a critical size for the phase transition. We suppose that the unique phonon dispersion and electronic structure, as well as the atomic-site fluctuation on the surface of the nanoparticle might play key roles, since the proportion of surface atoms increases with decreasing the particle size.

Correction in the manuscript:

On the contrary, for the ceria with larger particle size, the induced stress is less significant to soften the phonon due to the declined defect concentration, and thereby the phase transition is inhibited.

Q5: The conductivity can also be analyzed in terms of the electronic structure of the defective ceria.

A5: We have supplemented DFT calculation of spin charge density to verify the charged defects in the tetragonal ceria. Such localized electrons are reported to serve as mobile charge carrier, which could significantly increase the conductivity⁷.

Q6: The relevance of Figure 1c needs to be clarified.

A6: The thermal expansion curves (Fig. 1a-1b) clearly show distinct steps in the temperature range of $-25 - 75$ °C, indicating a structural break for nanosized ceria as temperature rises. In addition, our specific heat capacity measurement also indicates a phase transition that could be relevant to a local structural change. These results motivate us to directly study the atomic arrangement of the nanoparticles using STEM with annular bright-field (ABF) detector. The ABF image in the present study is a key evidence of the dispersively oxygen vacancies in nanosized ceria, which plays a pivotal role in the phonon softening and charge transfer. In order to strengthen the relevance of ABF-STEM image, we have changed the sequence of Fig. 1c and Fig. 1d, and also revised the related descriptions in the manuscript.

Correction in the manuscript:

In order to provide a direct view of the local structure, scanning transmission electron microscopy with annular bright-field detector (ABF-STEM) was carried out for 5 nm ceria (Fig. 1d). Along the [001] projection, individual cerium- and oxygen-atom columns appear alternately. Faint dark dots could be recognized at the oxygen sites, but the contrast of these dots fluctuates, which is firmly demonstrated by the inconsistent peak valleys of the oxygen intensity profile (Fig. 1d inset). This result indicates the dispersive oxygen vacancies distributed in the nanoparticles, which are expected to play key roles in the NTE and the phase transition observed in nanosized ceria.

Q7: The inset in Figure 1d needs to be redrawn to improve readability.

A7: We have redrawn the inset of Fig. 1d (Fig. 1c in the revised version), and an explanation has been also added in the figure caption.

Correction in the manuscript:

The entropy (ΔS) of this phase transition was estimated by integrating $(C_p - C_{fit})/T$, where C_{fit} is the background extracting with a polynomial function to $C_p(T)$ between -150 °C to 150 °C (Fig. 1c inset).

Correction in Figure 1:

Figure 1. Anomalous thermal expansion results from phase transition in nanosized ceria. (a). Temperature dependence of the lattice parameters extracting from XRD refinements for the CeO_2 particles in different sizes. The inset depicts the unit cell of the CeO_2 with the space group $Fm\bar{3}m$. The standard deviations on the

lattice constants obtained from Rietveld refinements are much smaller than the symbol size in the figure. **(b)**. Comparison of the diffraction peaks (1 1 1) at different temperatures. The solid line depicts trend of the peak positions as the temperature changes. **(c)**. Specific heat capacity of the 5 nm ceria measured from -150 °C to 150 °C. The red line in the inset is the heat capacity peak in the transition region excluding the fitted background (i.e., $(C_p - C_{fit})$), and the green line shows the estimated entropy obtained by integrating $(C_p - C_{fit})/T$. **(d)**. The annular bright-field (ABF) image with a common tilt axis of [0 0 1] for 5 nm ceria. The inset shows the corresponding ABF in line profile acquired along the oxygen-atom columns. The arrow shows oxygen vacancy site marked with hollow block.

Q8: An extensive revision of the English is required.

A8: We have carefully revised many parts of the manuscript, and checked it in multiple proof readings. The English editing has been highlighted in blue.

Responses to Reviewer #3

Q1: The English is readable, however, it should be corrected before publication. As several authors reside in the United States, this should not be too difficult to achieve.

A1: Thank you for your suggestion. The whole manuscript has been polished by an English native speaker, and the English editing has been presented by blue highlights.

Q2: The supplemental information should contain some more details. E.g., what

exactly is the configuration for the "low temperature attachment" for the PXRD instrument? Is this a flat plate setup? If so - then variable temperature data ought to be collected with an internal standard to correct for sample height changes as a function of temperature. Note that this is especially important when dealing with small changes in lattice constants as in the current paper! The PDF data clearly show a shortening of nearest-neighbor O-O distances, but the absolute numbers could be affected considerably.

A2: Thank you for your constructive comment. The low temperature attachment of PW 3040-X'Pert Pro diffractometer (PANalytical) is a flat plate setup. The exact configuration has been supplemented in Fig. S4. To ensure the accuracy and stability of temperature controlling, we had taken a series measures. The heat conducting grease was smeared between heating (cooling) stage and sample cell. All the XRD measurements were carried out under vacuum to reduce thermal fluctuation. Besides, the thermocouple is made of Pt100, which is considered reliable in the conducted temperature range. On the basis of the reviewer's advice, we have revisited our variable temperature data of 5 nm ceria involving quartz (SiO_2) internal standard to calibrate the lattice constants of nanosized ceria. The Rietveld refinements were carried out by fixing the SiO_2 lattice constants, obtained from linear regression of the quartz thermal expansion⁸ (Fig. R2a). Remarkably, we found that the difference of the lattice constants extracted with or without the internal standard is subtle (Fig. R2b and Table R2), which doesn't affect the final conclusion of the NTE and the phase transition. This is because the systematic errors associated with the thermal expansion

of the cryostat could be maximally eliminated through correcting zero-shift. Taken together, we believe that the lattice parameters obtained from XRD in the present study are reliable. Some supplements have been added in Experimental Section in Supplementary Materials.

Figure R2. (a) Example of the XRD Rietveld refinement for the 5 nm ceria with the internal standard of SiO₂. (b) Comparison of the lattice constants extracted from XRD with or without internal standard of SiO₂.

Table R2. Temperature dependence of lattice constants extracted from XRD refinement with and without the internal standard of quartz.

Temperature (°C)	Without SiO ₂	With SiO ₂
-150	5.40772 (4)	5.40894 (6)
-100	5.40966 (4)	5.41124 (6)
-50	5.41132 (4)	5.41280 (7)
-25	5.41255 (4)	5.41354 (7)

0	5.41179 (4)	5.41297 (7)
25	5.41165 (4)	5.41231 (6)
50	5.41111 (4)	5.41182 (7)
75	5.40987 (4)	5.41108 (7)
100	5.41017 (4)	5.41166 (7)
150	5.41242 (5)	5.41388 (7)
200	5.41401 (5)	5.41519 (7)
250	5.41583 (5)	5.41708 (7)
300	5.41816 (5)	5.41930 (7)
350	5.41986 (5)	5.42120 (7)

Added in Supplementary Materials:

The configuration of the low temperature attachment is shown in Fig. S4. The heating conducting grease was smeared between the stage and sample cell, and the thermocouple is made of Pt100.

The reliability of the obtained lattice constants were examined by an internal standard of SiO₂. The difference of the lattice constants extracted with or without the internal standard is subtle, which could be due to the zero-shift correction that maximally eliminating the systematic errors.

Figure S4. The configuration of the low temperature attachment of PW 3040-X'Pert Pro diffractometer.

Q3: It is unclear whether the reported materials actually adopt a single phase, or whether a 2-phase mixture is formed.

A3: Thank you for pointing out the unclear description in the manuscript. Before we response the comments concerning the nanostructure of ceria, two key issues needs to be clarified:

First, the XRD method was used to determine the long-range structure of ceria in the present study. For the bulk material, the crystal structure can be determined by this conventional Bragg diffraction experiments with high precision. However, such method cannot give the short-range nanostructure in the case of nanomaterial, because of the broadening and overlapping of the Bragg peaks at high angle⁹. Only the average lattice constants and a distinct NTE could be surely determined for the present 5 nm ceria. Consequently, neutron PDF technique, which is emerging as a powerful tool for studying the nanostructure¹⁰, has been carried out to reveal the local structure (especially the oxygen displacement) of nanosized ceria.

Second, the tetragonal structure ($P4_2/nmc$) observed in nano-ceria is a subgroup of the cubic structure ($Fm-3m$). For the tetragonal phase, three structural parameters are variable, i.e., the lattice constants (a_t, c_t) and the atomic coordinate of O along c -axis (O_z). When the conditions of $c_t = \sqrt{2} a_t$ and $O_z = 0.75$ are satisfied, the tetragonal structure turns into the cubic-fluorite structure (see the tetragonal-cubic relation in Fig. S7). Consequently, for both XRD and PDF refinements, the data that is fittable with the $Fm-3m$ model could be certainly described well by the $P4_2/nmc$ model, since the latter contains more variables than the former. In such case, the structure with higher symmetry should be used. Analogously, if the two-phase model, which contains more repeating variables, for XRD/PDF patterns cannot improve the refinements significantly, the structure should be regarded as single phase within the resolution of the structural-characterization methods.

In the present study, the PDF data of 5 nm ceria can be fitted well with the single-phase cubic model above 25 °C and the two-phase model cannot give a better description of the data (see the response in Q5). In addition, the PDF data below - 75 °C can be well fitted with the single-phase tetragonal model but cannot be well fitted by the cubic model. Based on the above rules, the nanosized ceria possesses single cubic structure at high temperature, and single tetragonal structure at low temperature. The tetragonal-cubic phase transition occurs in the intermediate temperature range from -25 °C to 75°C.

Q4: The authors clearly state that data up to 75 C require use of the tetragonal phase

to get a good fit to the PDF data, but do not discuss whether all data can be accounted for exclusively by the tetragonal phase (e.g., the X-ray data should also be fittable with this phase!).

A4: We appreciate the question mentioned in this comment. In the present study, all the XRD patterns could be fitted very well with the cubic model, as indicated by the low values of Chi^2 (R_p , R_{wp} and Chi^2 have been supplemented in Fig S5). We attempted to refine the XRD pattern of 5 nm ceria (collected at -100 °C) with the $P4_2/nmc$ structure. The refined patterns and the obtained results using cubic and tetragonal model are presented in Fig. R3 and Table R3, respectively. We found that the tetragonal model didn't improve the refinement significantly, and the c_t is approximately equal to $\sqrt{2} a_t$ (here, $\frac{c_t}{\sqrt{2} a_t} = 1.0003$, while $\frac{c_t}{\sqrt{2} a_t} = 1.006$ for the tetragonal structure from the corresponding PDF result), which means the obtained tetragonal structure is very close the cubic structure. This is because the peak broadening, which is generally observed for nanosized materials, conceals the peak splitting. In addition, the atomic coordinate cannot be exactly extracted for the very small nanoparticles through XRD, on account of the overlapped peaks especially at high angle^{11, 12}. In short, the tetragonal phase is difficult to be distinguished by XRD method due to its limited resolution, and only an NTE could be surely determined for the nanosized ceria. In the manuscript, all the XRD results (Fig. 1a, Fig. S5, Fig. S9 and Table S1) are based on the cubic-fluorite model.

The neutron PDF data can be fitted well with the cubic-fluorite model above 75 °C, but this cubic structure cannot be well reconciled below 75 °C, especially below -25

°C (see the increased agree-factor, R_w , as temperature decreases in Fig. S6). The tetragonal structure ($P4_2/nmc$) was found from the subgroups of the fluorite structure to give the best description of the data at low temperature. Of course, as mentioned above, the tetragonal subgroup with lower symmetry could also be suitable for the data above 75 °C. Actually, all the PDF results over the entire temperature range are given within tetragonal model in the manuscript, in order to obtain continuous structural evolution. When $c_t \approx \sqrt{2} a_t$ (see Fig. 2c) and $O_z \approx 0.75$ (Fig. S8a and Table S2), the nanosized ceria could be regarded as the cubic phase. On the contrary, the tetragonal structure could be recognized from the conditions of $c_t > \sqrt{2} a_t$ and $O_z > 0.75$. From the overall neutron PDF results, the conclusion could be drawn that the nanosized ceria adopts cubic structure at high temperature, and possesses tetragonal structure at low temperature.

Figure R3. The refinement patterns of 5 nm ceria using (a) cubic model and (b) tetragonal model.

Table R3. The lattice constants of 5 nm ceria and the corresponding Chi^2 extracted

from XRD refinements using $Fm-3m$ and $P4_2/nmc$ models.

$Fm-3m$	a_c (Å)	--	Chi^2
	5.40966 (4)	--	1.15
$P4_2/nmc$	$\sqrt{2} a_t$ (Å)	c_t (Å)	Chi^2
	5.40907 (4)	5.41070 (4)	1.15

Corrections in the manuscript:

The cubic-fluorite model was initially utilized for the Rietveld refinements of X-ray diffraction (XRD) to determine the thermal lattice evolution (Fig. S4-5).

The observed NTE indicates a structural break for the nanosized ceria, which, however, cannot be further determined by XRD. This is due to the broadening and overlapping of the Bragg peaks that significantly reduces the XRD resolution.

The low-r neutron PDF data (from 1.5 to 15 Å) of 5 nm ceria was initially adopted with the cubic-fluorite model. This fluorite model fits well to the data above 75 °C, but fails to describe the data below 75 °C, especially below -25 °C (see the increased agree-factor, R_w , as the temperature decreases from 75 °C to -25 °C, Fig. S6).

The PDF fitting was carried out with $P4_2/nmc$ model over the entire temperature range to obtain continuous structural evolution (Table S2).

Question 5: Figure 1a shows lattice constants for a cubic phase extracted over the entire temperature range, and so does Table S1, implying that the cubic phase is observed at all temperatures. Table S2 seems to imply that the tetragonal phase is

present at all temperatures up to 225 C. If both are correct/meaningful, then this should mean 2-phase coexistence. Of course, application of both models to the same data would then be necessary to determine relative amounts.

A5: As mentioned in Q4, all the XRD data were refined with the cubic model (Table S1), and all the PDF data were fitted with the tetragonal model (Table S2). Such distinction doesn't imply a two-phase coexistence.

By analyzing the PDF data, the 5 nm ceria could be cautiously regarded as a single tetragonal phase at low temperature. The two-phase fitting with both cubic and tetragonal models were carried out for PDF data at -150 °C (Fig. R4). Such fitting didn't show a significant improvement in contrast with the single-phase fitting, and the cubic phase has a very limited relative amount (5.69%). This low content of cubic ceria might be an illusion, since the additional cubic model involves more variable parameters that would tolerate the mismatch of the noise signal from Fourier transform at high Q of $F(Q)$. From the experimental PDF point of view, the nanostructure of 5 nm ceria could be described as a single tetragonal phase at low temperature.

Figure R4. The comparison of the PDF fittings with (a) two phase and (b) single phase models.

Correction in the manuscript:

It is worth mentioning that the PDF refinement didn't show a significant improvement with two-phase fitting with both cubic and tetragonal models. As a consequence, the 5 nm ceria should be regarded as single tetragonal structure at low temperature.

Question 6: Figure 2c shows tetragonal lattice constants at low temperature, and I presume cubic ones at higher T (this should be indicated somewhere!) - but the values for the tetragonal a parameters do not agree with the numbers in Table S2.

A6: Thank you for pointing out the misleading part. The lattice constants in Fig. 2c was extracted from the PDF data with the tetragonal model. The unit cell of this

tetragonal structure is smaller than that of the cubic structure (Fig. S7). In order to keep the coherence of the thermal evolution along a -axis, $\sqrt{2} a_t$ (a_t is the a -parameter in Table S2) was utilized in Fig. 2c. We have made the descriptions more explicit in the main text, and a column of $\sqrt{2} a_t$ has been added in Table S2

Correction in the manuscript:

This tetragonal structure can be regarded as the shear strain of the oxygen sublattice distorted along fourfold axis (Fig. 2c inset), which leads to a smaller tetragonal unit cell than the cubic one. Based on the conversion relation between $Fm-3m$ and $P4_2/nmc$ (Fig. S7), the obtained tetragonal structure (a_t, c_t) can be expanded to a cubic-like unit cell ($a = \sqrt{2} a_t, c = c_t$). This expanded unit cell has been adopted in the present study to describe the tetragonal phase, so as to unify its lattice constants towards the cubic structure.

Correction in Table S2:

Table S2.

Lattice constants of 5 nm ceria extracted from the low-r nPDF refinement.

T (°C)	a_t (Å)	$\sqrt{2} a_t$	c (Å)	O_z	R_w (%)
-150	3.8175 (30)	5.3987 (42)	5.4260 (52)	0.7578 (9)	9.9
-100	3.8184 (32)	5.4001 (45)	5.4323 (58)	0.7571 (9)	10.1
-50	3.8204 (31)	5.4029 (44)	5.4323 (54)	0.7572 (10)	10.3
-25	3.8210 (33)	5.4038 (47)	5.4348 (60)	0.7573 (9)	10.4

0	3.8215 (38)	5.4045 (54)	5.4299 (72)	0.7554 (15)	11.2
25	3.8228 (34)	5.4062 (48)	5.4218 (62)	0.7543 (13)	10.8
50	3.8246 (29)	5.4088 (41)	5.4147 (50)	0.7518 (12)	10.6
75	3.8259 (35)	5.4106 (49)	5.4082 (66)	0.7501 (13)	10.8
125	3.8271 (36)	5.4123 (51)	5.4107 (68)	0.7497 (13)	10.8
225	3.8302 (38)	5.4167 (54)	5.4159 (72)	0.7493 (11)	10.7

Question 7: Figure S8 claims to be displaying the a-parameter - yet once again, it is unclear whether this is the cubic or tetragonal parameter.

A7: The lattice parameters in Fig. S8 (Fig. S9 in the revised version) were extracted from XRD Rietveld refinements. As mentioned in Q4, the tetragonal phase can't be distinguished by XRD method. Nevertheless, the reversibility of the phase transition could be surely determined by the distinct step of the thermal expansion under heating/cooling cycles.

Question 8: Similarly, is Figure S6 displaying both cubic and tetragonal volumes? This is rather confusing, and makes it hard to truly evaluate the results presented.

A8: In order to make the volumes of tetragonal and cubic phase comparable, an expanded unit cell, whose volume is two times larger (see Fig. R5), was utilized for the tetragonal phase in Fig. S6b (Fig. S8b in the revised version). This leads to a better understanding of the volume contraction during the tetragonal-cubic phase transition. The volumes of tetragonal ceria are presented below -25 °C, while the

volumes of cubic ceria are presented above 75 °C. In the intermediate temperature range, a volumetric contraction occurs along with the tetragonal-cubic phase transition. We have re-edited the corresponding figure (Fig. S8), and some explanation has also been added in to avoid the misunderstanding.

Figure R5. Schematic diagram of the unit cell used in Figure S6b expanded from the tetragonal unit cell.

Correction in the manuscript:

During the tetragonal-cubic phase transition from -25 °C to 75 °C, c -axis contracts towards the length of a -axis ($\sqrt{2} a_t$), while the oxygen coordinate along c -axis (O_z) approaches to 0.75 (Fig. S8a). This makes a volume contraction of 1.3 ‰ excluding the general thermal expansion (Fig. S8b).

Re-edited Fig. S8:

Figure S8. Detail results of the nPDF refinements. (a). Relative atomic coordinates of oxygen sites as a function of temperature. **(b).** Volume of the unit cell as a function of temperature.

Added in Supplementary Materials:

In Fig. S8b, an expanded unit cell, whose volume is two times larger than the original $P4_2/nmc$ structure, has been utilized for the tetragonal phase at low temperature. This makes the volumes of the tetragonal and cubic phase comparable. From the PDF results, a volume contraction is observed during the phase transition, which is consistent with the XRD result.

Question 9: The authors need to clarify which phase is present at what temperature and refine/fit ALL data accordingly. E.g., if the material is not cubic at low T - then

plotting a cubic lattice parameter as $f(T)$ at low T is not meaningful. If the material forms a 2-phase mixture, obviously major revisions will be necessary to adequately present the results.

A9: We appreciate the above questions concerning the structural characterization. Accordingly, we have clarified the nanostructure of ceria explicitly, and revised the unclear descriptions regarding to the XRD and the PDF analysis in the manuscript. Although the XRD method cannot distinguish the tetragonal structure from the cubic one, we suppose it meaningful to display the lattice parameters extracted from XRD as a function of temperature, because this provides a direct view of the phase transition through a distinct NTE.

Reference:

- [1] Keating, P. R., Scanlon, D. O., Morgan, B. J., Galea, N. M. & Watson, G. W. Analysis of intrinsic defects in CeO₂ using a Koopmans-like GGA+ U approach. *J. Phys. Chem. C* **116**, 2443-2452 (2012).
- [2] Budai, J. D. *et al.* Metallization of vanadium dioxide driven by large phonon entropy. *Nature* **515**, 535-539 (2014).
- [3] Li, H. Y. *et al.* Multiple configurations of the two excess 4*f* electrons on defective CeO₂ (1 1 1): Origin and implications. *Phys. Rev. B* **79**, (2009).
- [4] Greenberg, M., Wachtel, E., Lubomirsky, I., Fleig, J. & Maier, J. Elasticity of solids with a large concentration of point defects. *Adv. Funct. Mater.* **16**, 48-52

(2006).

[5] Atkinson, A. Chemically-induced stresses in gadolinium-doped ceria solid oxide fuel cell electrolytes. *Solid State Ionics* **95**, 249-258 (1997).

[6] Kümmerle, E. & Heger, G. The Structures of $C-Ce_2O_{3+\delta}$, Ce_7O_{12} , and $Ce_{11}O_{20}$. *J. Solid State Chem.* **147**, 485-500 (1999).

[7] Rojac, T. *et al.* Domain-wall conduction in ferroelectric $BiFeO_3$ controlled by accumulation of charged defects. *Nature Mater.* (2016).

[8] Rammo, N. N. & Farid, S. B. Thermal expansion coefficients and Gruneisen parameters of quartz at high temperature by X-ray method. *Powder Diffr.* **9**, 148-150 (1994).

[9] Young, C. A. & Goodwin, A. L. Applications of pair distribution function methods to contemporary problems in materials chemistry. *J. Mater. Chem.* **21**, 6464-6476 (2011).

[10] Zobel, M., Neder, R. B. & Kimber, S. A. Universal solvent restructuring induced by colloidal nanoparticles. *Science* **347**, 292-294 (2015).

[11] Wang, J. *et al.* Negative-pressure-induced enhancement in a freestanding ferroelectric. *Nature Mater.* **14**, 985-990 (2015).

[12] Zhu, H. *et al.* Hydration and thermal expansion in anatase nanoparticles. *Adv. Mater.* **28**, 6894-6899 (2016).

Reviewers' comments:

Reviewer #2 (Remarks to the Author):

The authors have introduced many of my comments in previous correspondence yet I have still some points that deserve further explanations.

1) The U values need to be added to the main text.

2) The use of U in the O2p states although reported it is not of common use and besides some issues in the VASP implementation for this particular case are known. Therefore, more tests on the effect of the U in the calculated properties are needed, like the vacancy formation energy as a function of U. In addition, the English use in the SI needs revision. For instance, "The U value of Ce f states was set to 5 eV, and for O2p states, we employed a U value of 5.5 eV, as suggested in the previous theoretic works"

3) The authors consider that 36 calculations in the 2x2x2 is a high computational burden. Actually with the present computers it is not and as I consider that this particular result is important I recommend again the authors to address my previous Question 2 in a more adequate manner.

4) Figure S14 shows no localized electrons in a single Ce atom but rather the average structure with multiple Ce getting density by reduction. I refer the authors to the extensive literature describing that localization improves the energetics of the systems by significant amounts and thus that a more adequate treatment of the localization needs to be implemented.

Reviewer #3 (Remarks to the Author):

The authors have adequately addressed my comments and nicely clarified the open questions. I am only requesting one minor change: The difference in lattice constants with and without internal standard is NOT due to a "zero-shift error", it is due to a change in sample height (which is similar to zero error at low angles, but follows a different angular dependence). Thus it should be called a sample height error, not a zero shift.

Also, while the difference may be small, the authors should use the best possible numbers, which means the ones collected with internal standard. There is no need to use/report the uncorrected numbers!

Responses to Reviewers' comments

The point-by-point responses to the reviewers' comments are given below. All the corrections in the manuscript are highlighted in blue.

Responses to Reviewer #2

Q1: The U values need to be added to the main text.

A1: Thank you for the comment. The U values used in the present study have been added in the main text.

Correction in the manuscript:

The lattice dynamics, as well as the formation of intrinsic defect of nanosized ceria, has been investigated using density functional theory (DFT) with the generalized gradient approximation corrected for on-site Coulombic interactions (GGA+ U , Ref 1). The U values of Ce $_{4f}$ and O $_{2p}$ states were set to 5 eV and 5.5 eV, respectively, in order to overcome the self-interaction error (SIE) in the reduced CeO $_2$ system² (see the Computational Methods in Supplementary Information).

Q2: The use of U in the O $2p$ states although reported it is not of common use and besides some issues in the VASP implementation for this particular case are known. Therefore, more tests on the effect of the U in the calculated properties are needed, like the vacancy formation energy as a function of U . In addition, the English use in the SI needs revision. For instance,

"The U value of Ce f states was set to 5 eV, and for O $2p$ states, we employed a U

value of 5.5 eV, as suggested in the previous theoretic works"

A2: Thank you for the comment. It is known that standard LDA/GGA DFT functionals are incapable of correctly modeling defect states for a number of wide-gap oxides³⁻⁶, including the reduced CeO₂ system^{1, 2, 7, 8}. This is due to the self-interaction error (SIE) inherent to such functionals, leading to an artificial bias toward delocalization of partially occupied states⁹. When employing GGA (LDA) to describe O-derived defect systems, the above problem is acute not only for localized metal *d* and *f* states, but also for O *p* states^{3-5, 10-12}. The later failure has been traced back to a residual self-interaction present within O *p* shell in GGA or LDA¹³.

By applying GGA+*U* to the Ce_{4*f*} and O_{2*p*} states, an improved description of defects with localized electrons and holes could be obtained^{2, 12}. Previous theoretical studies demonstrated that the expected structure of reduced CeO₂ could be reproduced with the value of $U\{\text{Ce}_{4f}\} = 5.0 \text{ eV}^{14-17}$. As for the O_{2*p*} states, the suitable value of $U\{\text{O}_{2p}\}$ is $5.5 \text{ eV}^{2, 8, 12}$, determined from a Koopmans-like fitting procedure³. This method is based on the result that the energy change of a system when an electron is added or removed should be linear¹⁸. Compared with the results of $U_{\text{Ce, O}} = \{5.0, 0\} \text{ eV}$, the defect structure could be preferably modeled employing the derived value of $U_{\text{Ce, O}} = \{5.0, 5.5\} \text{ eV}$, leading to a reasonable value of vacancy formation energy (-9.67 eV)². Consequently, we set $U\{\text{O}_{2p}\} = 5.5 \text{ eV}$ in the present study to correct the SIE associated with the O_{2*p*} states.

As referee suggested, we performed some tests on the effects of $U\{\text{O}_{2p}\}$ on the calculated properties, like the vacancy formation energy as a function of $U\{\text{O}_{2p}\}$ (Fig.

R1a), as well as the phonon dispersions with and without $U\{O_{2p}\}$ (Fig. R1b). Consistent with the previous study², we found that the result of vacancy formation energy is more reasonable with $U\{O_{2p}\} = 5.5$ eV than that with $U\{O_{2p}\} = 0$. In addition, the distinction is insignificant between the phonon dispersions with and without $U\{O_{2p}\}$. As a consequence, we employ $U\{O_{2p}\} = 5.5$ eV to model the defect structure preferably with minimal effect on the other properties.

As suggested, the English usage has been revised, and more explanations with respect to the U values have been included in SI.

Figure R1: (a) Vacancy formation energies of both cubic and tetragonal unit cell (using the experimental PDF results at -150 °C and 200 °C, respectively) as a function of U_O . (b) Phonon dispersions of cubic ceria with $U\{O_{2p}\} = 0$ and $U\{O_{2p}\} = 5.5$ eV, respectively, with $U\{Ce_{4f}\} = 5$ eV.

Correction in SI:

All the calculations in the present study were performed with Vienna ab initio simulation package (VASP)¹⁹, using the Perdew-Burke-Ernzerhof (PBE) generalized gradient approximation (GGA) and the projector augmented wave (PAW) potential²⁰.

The energy cutoff of the plane-wave basis is 500 eV. It is known that standard DFT functionals are incapable of correctly modeling O-derived defect states due to the inherent self-interaction error (SIE)^{1-3, 5}. For the reduced CeO₂ system, i.e., CeO₂ that contains intrinsic O vacancies, such problem is acute for both Ce_{4f} and O_{2p} states^{2, 8, 12}. To correct the SIE associated with the vacancies, $U\{\text{Ce}_{4f}\} = 5.0$ eV was applied to the Ce 4f states¹⁴⁻¹⁷, while $U\{\text{O}_{2p}\} = 5.5$ eV, which is determined from a Koopmans-like fitting process³, was applied to the O 2p states^{2, 8, 12}.

Q3: The authors consider that 36 calculations in the 2x2x2 is a high computational burden. Actually with the present computers it is not and as I consider that this particular result is important I recommend again the authors to address my previous Question 2 in a more adequate manner.

A3: We appreciate the question mentioned in this comment, and the issue of concern has been deliberated carefully before we make the response.

First, we would like to make it clear that the $2 \times 2 \times 2$ supercell of CeO_{1.75} with different arrangements of oxygen vacancies will lead to 36 configurations (not 36 calculations, we apologize for the mistake in the previous response), most of them have *P1* space group based on the symmetry analysis (discussed in **Appendix R1**). For each configuration, such a low symmetry will give rise to hundreds of displacement patterns when we use the finite displacement method for the phonon dispersion calculations (i.e. hundreds of POSCAR files with Phonopy software and VASP calculation), so in total there will be over ten thousands POSCAR files for the

listed 36 configurations (**Table R1**). This is a tremendous computing workload if we calculate the phonon dispersions for all the configurations with different arrangement of oxygen vacancies.

In addition, our STEM-ABF image (Fig. 1d) firmly evidences that the oxygen vacancies are distributed dispersively without any ordering in the lattice of the nanoparticles. Therefore, it is not likely that some particular arrangement of the generated vacancies induces the tetragonal distortion. Thus, we consider the stress effect induced by the oxygen vacancies on the structure phase transition rather than the effect of vacancy formation. It is known that the heterogeneously distributed oxygen vacancies in ceria could give rise to the chemically induced stress^{21, 22}, which, according to our DFT results, leads to the phonon softening that drives the observed tetragonal-cubic phase transition.

Appendix R1:

The typical fluorite structure was applied to build the $2 \times 2 \times 2$ supercell of ceria (96 atoms). From the XPS results (Fig. S12), the concentration of the oxygen vacancies in 5 nm ceria is around 15 %. Consequently, eight vacancies have been included in the $2 \times 2 \times 2$ supercell (corresponding to $\text{CeO}_{1.75}$) in different configurations. We removed one oxygen vacancy for each unit cell (eight in total), and the vacancy sites are numbered from 1 to 8 (Fig. R2a). In addition, the locations of the eight unit cells are also marked from A to H (Fig. R2b). As a result, the location of each oxygen vacancy in the supercell could be marked, such as A1, F7, etc.

We distinguish the configurations by their specific vacancy arrangements (e.g.,

A1-B1-C1-D1-E1-F1-G1-H1). Because of the translational symmetry between the oxygen atoms in the unit cell, as well as the unit cells in the supercell, some configurations are equivalent (e.g., A1-B1-C1-D1-E1-F1-G1-H1 \leftrightarrow A2-B2-C2-D2-E2-F2-G2-H2). Analogously, the central symmetry of the supercell also leads to some equivalent configurations (e.g., A1-B2-C3-D4-E5-F6-G7-H8 \leftrightarrow A8-B7-C6-D5-E4-F3-G2-H1). So there are 36 configurations with different vacancy arrangements in the lattice, which have been listed in Table R1. The overall number of POSCAR files generated from the listed configurations is 12963.

Figure R2: Schematic diagram of (a) oxygen vacancy sites and (b) unit cell locations

Table R1.

The list of the configurations with different arrangements of oxygen vacancies in $\text{CeO}_{1.75}$ $2 \times 2 \times 2$ supercell.

No.	Configurations	Number of POSCARs	Space group	Space group number
1	A1-B1-C1-D1-E1-F1-G1-H1	5	P-43m	215
2	A2-B1-C1-D1-E1-F1-G1-H1	150	Cmm2	35

3	A3-B1-C1-D1-E1-F1-G1-H1	150	Cmm2	35
4	A4-B1-C1-D1-E1-F1-G1-H1	284	Cm	8
5	A5-B1-C1-D1-E1-F1-G1-H1	150	Cmm2	35
6	A6-B1-C1-D1-E1-F1-G1-H1	284	Cm	8
7	A7-B1-C1-D1-E1-F1-G1-H1	284	Cm	8
8	A8-B1-C1-D1-E1-F1-G1-H1	108	R3m	160
9	A2-B2-C1-D1-E1-F1-G1-H1	38	P-42m	111
10	A3-B2-C1-D1-E1-F1-G1-H1	528	P1	1
11	A4-B2-C1-D1-E1-F1-G1-H1	264	P2	3
12	A5-B2-C1-D1-E1-F1-G1-H1	528	P1	1
13	A6-B2-C1-D1-E1-F1-G1-H1	264	P2	3
14	A7-B2-C1-D1-E1-F1-G1-H1	284	Cm	8
15	A8-B2-C1-D1-E1-F1-G1-H1	284	Cm	8
16	A3-B3-C2-D1-E1-F1-G1-H1	528	P1	1
17	A4-B3-C2-D1-E1-F1-G1-H1	284	Cm	8
18	A5-B3-C2-D1-E1-F1-G1-H1	284	Cm	8
19	A6-B3-C2-D1-E1-F1-G1-H1	528	P1	1
20	A7-B3-C2-D1-E1-F1-G1-H1	528	P1	1
21	A8-B3-C2-D1-E1-F1-G1-H1	284	Cm	8
22	A4-B4-C3-D2-E1-F1-G1-H1	528	P1	1
23	A5-B4-C3-D2-E1-F1-G1-H1	528	P1	1

24	A6-B4-C3-D2-E1-F1-G1-H1	528	P1	1
25	A7-B4-C3-D2-E1-F1-G1-H1	528	P1	1
26	A8-B4-C3-D2-E1-F1-G1-H1	528	P1	1
27	A5-B5-C4-D3-E2-F1-G1-H1	528	P1	1
28	A6-B5-C4-D3-E2-F1-G1-H1	528	P1	1
29	A7-B5-C4-D3-E2-F1-G1-H1	528	P1	1
30	A8-B5-C4-D3-E2-F1-G1-H1	528	P1	1
31	A6-B6-C5-D4-E3-F2-G1-H1	528	P1	1
32	A7-B6-C5-D4-E3-F2-G1-H1	528	P1	1
33	A8-B6-C5-D4-E3-F2-G1-H1	528	P1	1
34	A7-B7-C6-D5-E4-F3-G2-H1	284	Cm	8
35	A8-B7-C6-D5-E4-F3-G2-H1	18	Pm-3m	221
36	A8-B8-C6-D5-E4-F3-G2-H1	284	Cm	8

Q4: Figure S14 shows no localized electrons in a single Ce atom but rather the average structure with multiple Ce getting density by reduction. I refer the authors to the extensive literature describing that localization improves the energetics of the systems by significant amounts and thus that a more adequate treatment of the localization needs to be implemented.

A4: Thanks the referee for the comment. In the present study, electron paramagnetic resonance (EPR) was employed to directly observe the paramagnetic centers, i.e., excess spins with unpaired electrons, in both tetragonal and cubic ceria. From the

EPR results (Fig. 4a-b), excess spins in both oxygen vacancies and Ce atoms (resulting in Ce^{3+} states) were found in tetragonal ceria, whereas only Ce^{3+} states were found in cubic ceria.

On the other hand, the above charge transition between oxygen vacancies and Ce (III) states was also verified by the DFT calculation of spin charge density, which has been frequently used to describe the electronic structure associated with defects^{5, 23}. The tetragonal and cubic unit cells were constructed with the experimental PDF results (at -150 °C and 200 °C, respectively) before removing one oxygen atom. Remarkably, excess spins were found in the vacancy of tetragonal lattice, while no excess spins were found in the vacancy of cubic lattice. The calculated results (Fig.S14) accord well with the EPR experiment, and we suppose the distinction of charge states could play a key role in the stability of different configurations.

Upon oxygen removal, Ce atoms around vacancies could be reduced by the residual electrons. The concentration of oxygen vacancies is about 15 % according to the XPS, so we removed one oxygen atom per unit cell (12.5 % oxygen vacancies, $\text{CeO}_{1.75}$) for the calculation. In such case, all the Ce atoms will neighbor one oxygen vacancy on average (considering the equivalent sites of Ce sites), so multiple Ce atoms get spin density from the oxygen vacancies. In other words, the large amount of oxygen vacancies leads to the reduction of multiple Ce atoms in both tetragonal and cubic structure.

Correction in the manuscript:

As seen in Fig. 4a, the symmetric signal at $g = 2.003$ is assigned to the unpaired

electrons trapped in oxygen vacancies (paramagnetic defect with excess spins, V_o^-)²⁴, whereas the axial signals with $g_{\perp} = 1.967$ and $g_{\parallel} = 1.947$ are assigned to the paramagnetic Ce^{3+} sites with unpaired f electrons²⁵. During the phase transition upon heating, a charge transfer occurs from the (V_o^- s) of tetragonal ceria to the Ce f orbitals of cubic ceria (Fig. 4b), suggesting that the charge states of defects in the energy gap could play a key role in the stability of different configurations. In addition, such charge transfer between tetragonal and cubic phase has been also verified by our DFT calculation of spin charge density (Fig. S14, details in in Supplementary Materials).

Correction in SI:

Notably, excess spins were found in the vacancy of tetragonal lattice, while no excess spins were found in the vacancy of cubic lattice. The calculated results accord well with the EPR experiment (Fig. 4a-b), and we suppose the distinction of charge states could play a key role in the stability of different configurations

Responses to Reviewer #3

Q1: The difference in lattice constants with and without internal standard is NOT due to a "zero-shift error", it is due to a change in sample height (which is similar to zero error at low angles, but follows a different angular dependence). Thus it should be called a sample height error, not a zero shift.

A1: Thank you for the correction. The inaccurate description in SI has been revised.

Correction in SI:

Note that the difference of the lattice constants extracted with or without the internal standard is subtle, which indicates the systematic error, derived from the change in sample height, has been corrected maximally through the Rietveld refinements.

Q2: Also, while the difference may be small, the authors should use the best possible numbers, which means the ones collected with internal standard. There is no need to use/report the uncorrected numbers!

A2: Thank you for the suggestion. We have included the lattice constants of 5 nm ceria calibrated with internal standard in the present study.

Correction in Figure 1:

Figure 1. Anomalous thermal expansion results from phase transition in

nanosized ceria. (a). Temperature dependence of the lattice parameters extracting from XRD refinements for the CeO₂ particles in different sizes. The inset depicts the unit cell of the CeO₂ with the space group *Fm-3m*. The standard deviations on the lattice constants obtained from Rietveld refinements are much smaller than the symbol size in the figure. **(b).** Comparison of the diffraction peaks (1 1 1) at different temperatures. The solid line depicts trend of the peak positions as the temperature changes. **(c).** Specific heat capacity of the 5 nm ceria measured from -150 °C to 150 °C. The red line in the inset is the heat capacity peak in the transition region excluding the fitted background (i.e., $(C_p - C_{fit})$), and the green line shows the estimated entropy obtained by integrating $(C_p - C_{fit})/T$. **(d).** The annular bright-field (ABF) image with a common tilt axis of [0 0 1] for 5 nm ceria. The inset shows the corresponding ABF in line profile acquired along the oxygen-atom columns. The arrow shows oxygen vacancy site marked with hollow block.

Correction in SI:

For the 5 nm ceria, the lattice constants extracted from variable temperature data have been calibrated by quartz (SiO₂) internal standard (Fig. S5a).

Correction in Figure S5:

Figure S5. Examples of Rietveld refinement for the samples of (a) 5 nm, (b) 9 nm, (c) 18 nm and (d) bulk taken at room temperature. The quartz internal standard has been mixed with the 5 nm ceria for calibration. The raw experimental data are shown with the red hollow circles. The black lines are the calculated results based on the *Fm-3m* space group. The green vertical lines show the peak positions. The blue lines show the difference between the raw data and the calculated patterns.

Correction in Table S1:

Table S1.

The unit cell parameters (*a*-axis) at different temperatures for 5 nm, 9 nm, 18 nm and bulk ceria obtained using XRD Rietveld refinement.

T (°C)	5 nm (Å)	9 nm(Å)	18 nm (Å)	bulk (Å)
-150	5.40894 (6)	----	5.40595 (8)	5.40377 (4)
-135	----	5.40668 (4)	----	----
-100	5.41124 (6)	5.40872 (4)	5.40751 (8)	5.40526 (4)
-50	5.41280 (7)	5.41068 (4)	5.40991 (8)	5.40711 (4)
-25	5.41354 (7)	----	----	----
0	5.41297 (7)	5.41141 (4)	5.41072 (6)	5.40914 (4)
25	5.41231 (6)	5.41138 (4)	5.41118 (6)	4.41049 (4)
50	5.41182 (7)	5.41124 (5)	5.41198 (6)	5.41184 (4)
75	5.41108 (7)	5.41154 (5)	5.41240 (6)	5.41311 (4)
100	5.41166 (7)	5.41204 (5)	5.41332 (6)	5.41468 (3)
150	5.41388 (7)	5.41329 (5)	5.41527 (6)	5.41758 (3)
200	5.41519 (7)	5.41495 (5)	5.41716 (6)	5.42042 (3)
250	5.41708 (7)	5.41707 (5)	5.41932 (6)	5.42351 (3)
300	5.41930 (7)	5.41917 (5)	5.42219 (6)	5.42657 (3)
350	5.42120 (7)	5.42193 (5)	5.42530 (6)	5.42959 (3)

Reference:

1. Paier, J., Penschke, C. & Sauer, J. Oxygen defects and surface chemistry of ceria: quantum chemical studies compared to experiment. *Chem. Rev.* **113**, 3949-3985 (2013).
2. Keating, P. R., Scanlon, D. O., Morgan, B. J., Galea, N. M. & Watson, G. W. Analysis of intrinsic defects in CeO₂ using a Koopmans-like GGA+ *U* approach. *J. Phys. Chem. C* **116**, 2443-2452 (2012).
3. Lany, S. & Zunger, A. Polaronic hole localization and multiple hole binding of acceptors in oxide wide-gap semiconductors. *Phys. Rev. B* **80**, 085202 (2009).
4. Scanlon, D. O. *et al.* Surface sensitivity in lithium-doping of MgO: a density functional theory study with correction for on-site Coulomb interactions. *J. Phys. Chem. C* **111**, 7971-7979 (2007).
5. Morgan, B. J. & Watson, G. W. Polaronic trapping of electrons and holes by native defects in anatase TiO₂. *Phys. Rev. B* **80**, 233102 (2009).
6. Scanlon, D. O., Morgan, B. J., Watson, G. W. & Walsh, A. Acceptor levels in *p*-type Cu₂O: rationalizing theory and experiment. *Phys. Rev. Lett.* **103**, 096405 (2009).
7. Plata, J. J., Márquez, A. M. & Sanz, J. F. Improving the density functional theory + *U* description of CeO₂ by including the contribution of the O 2*p* electrons. *J. Chem. Phys.* **2012**, 136 (4), 041101.
8. Lucid, A. K., Keating, P. R., Allen, J. P. & Watson, G. W. Structure and

- reducibility of CeO₂ doped with trivalent cations. *J. Phys. Chem. C* **120**, 23430-23440 (2016).
9. Mori-Sánchez, P., Cohen, A. J. & Yang, W. Localization and delocalization errors in density functional theory and implications for band-gap prediction. *Phys. Rev. Lett.* **100**, 146401 (2008).
10. Nolan, M. & Watson, G. W. Hole localization in Al doped silica: A DFT + *U* description. *J. Chem. Phys.* **125**, 144701 (2006).
11. Morgan, B. J. & Watson, G. W. Intrinsic n-type defect formation in TiO₂: a comparison of rutile and anatase from GGA + *U* calculations. *J. Phys. Chem. C* **114**, 2321-2328 (2010).
12. Keating, P. R., Scanlon, D. O. & Watson, G. W. Computational testing of trivalent dopants in CeO₂ for improved high- κ dielectric behaviour. *J. Mater. Chem. C* **1**, 1093-1098 (2013).
13. Lægsgaard, J. & Stokbro, K. Hole trapping at Al impurities in silica: A challenge for density functional theories. *Phys. Rev. Lett.* **86**, 2834 (2001).
14. Nolan, M., Grigoleit, S., Sayle, D. C., Parker, S. C. & Watson, G. W. Density functional theory studies of the structure and electronic structure of pure and defective low index surfaces of ceria. *Surf. Sci.* **576**, 217-229 (2005).
15. Scanlon, D. O., Galea, N. M., Morgan, B. J. & Watson, G. W. Reactivity on the (110) surface of ceria: A GGA + *U* study of surface reduction and the adsorption of CO and NO₂. *J. Phys. Chem. C* **113**, 11095-11103 (2009).

16. Zhang, C., Michaelides, A., King, D. A. & Jenkins, S. J. Structure of gold atoms on stoichiometric and defective ceria surfaces. *J. Chem. Phys.* **129**, 194708 (2008).
17. Andersson, D. A., Simak, S., Johansson, B., Abrikosov, I. & Skorodumova, N. V. Modeling of CeO₂, Ce₂O₃, and CeO_{2-x} in the LDA + *U* formalism. *Phys. Rev. B* **75**, 035109 (2007).
18. Perdew, J. P. *et al.* Exchange and correlation in open systems of fluctuating electron number. *Phys. Rev. A* **76**, 040501 (2007).
19. Kresse, G. & Furthmüller, J. Efficiency of ab-initio total energy calculations for metals and semiconductors using a plane-wave basis set. *Comput. Mater. Sci.* **6**, 15-50 (1996).
20. Kresse, G. & Furthmüller, J. Efficient iterative schemes for ab initio total-energy calculations using a plane-wave basis set. *Phys. Rev. B* **54**, 11169 (1996).
21. Greenberg, M., Wachtel, E., Lubomirsky, I., Fleig, J. & Maier, J. Elasticity of solids with a large concentration of point defects. *Adv. Funct. Mater.* **16**, 48-52 (2006).
22. Atkinson, A. Chemically-induced stresses in gadolinium-doped ceria solid oxide fuel cell electrolytes. *Solid State Ionics* **95**, 249-258 (1997).
23. Li, H.-Y. *et al.* Multiple configurations of the two excess 4*f* electrons on defective CeO₂ (1 1 1): Origin and implications. *Phys. Rev. B* **79**, 193401 (2009).
24. Gionco, C. *et al.* Paramagnetic defects in polycrystalline zirconia: An EPR and DFT study. *Chem. Mater.* **25**, 2243-2253 (2013).

25. Abi-aad, E., Bechara, R., Grimblot, J. & Aboukais, A. Preparation and characterization of ceria under an oxidizing atmosphere. Thermal analysis, XPS, and EPR study. *Chem. Mater.* **5**, 793-797 (1993).

REVIEWERS' COMMENTS:

Reviewer #2 (Remarks to the Author):

The authors have successfully incorporated all the comments I had to the previous version. Therefore, I can recommend the manuscript for publication in its present form.

Reviewer #3 (Remarks to the Author):

The manuscript can be published as is, as the authors have adequately addressed my comments.